# Dynamics of the Lipidome in a Colon Simulator

**DOI:** 10.3390/metabo13030355

**Published:** 2023-02-27

**Authors:** Matilda Kråkström, Alex M. Dickens, Marina Amaral Alves, Sofia D. Forssten, Arthur C. Ouwehand, Tuulia Hyötyläinen, Matej Orešič, Santosh Lamichhane

**Affiliations:** 1Turku Bioscience Centre, University of Turku and Åbo Akademi University, 20520 Turku, Finland; 2Global Health and Nutrition Sciences, International Flavors & Fragrances, 02460 Kantvik, Finland; 3School of Science and Technology, Örebro University, 702 81 Örebro, Sweden; 4School of Medical Sciences, Faculty of Medicine and Health, Örebro University, 702 81 Örebro, Sweden

**Keywords:** lipidomics, metabolomics, gut microbiome, in vitro colon simulator

## Abstract

Current evidence suggests that gut microbiome-derived lipids play a crucial role in the regulation of host lipid metabolism. However, not much is known about the dynamics of gut microbial lipids within the distinct gut biogeographic. Here we applied targeted and untargeted lipidomics to in vitro-derived feces. Simulated intestinal chyme was collected from in vitro gut vessels (V1–V4), representing proximal to distal parts of the colon after 24 and 48 h with/without polydextrose treatment. In total, 44 simulated chyme samples were collected from the in vitro colon simulator. Factor analysis showed that vessel and time had the strongest impact on the simulated intestinal chyme lipid profiles. We found that levels of phosphatidylcholines, sphingomyelins, triacylglycerols, and endocannabinoids were altered in at least one vessel (V1–V4) during simulation. We also found that concentrations of triacylglycerols, diacylglycerols, and endocannabinoids changed with time (24 vs. 48 h of simulation). Together, we found that the simulated intestinal chyme revealed a wide range of lipids that remained altered in different compartments of the human colon model over time.

## 1. Introduction

The human gut harbors trillions of microbes that exhibit a mutually beneficial relationship with the host [1]. A key contribution of the gut microbiota to the host is nutrient and xenobiotic metabolism, which plays a major role in training the immune system and promoting intestinal homeostasis [2,3]. Moreover, gut microbes are essential for the maintenance of the host’s metabolic homeostasis. Specific disturbances in the gut microbiome composition may contribute to a wide range of diseases, including inflammatory bowel disease [4], nonalcoholic liver disease [5], and psychiatric disorders [6]. Identifying the compositional changes in the gut microbiome alone, however, does not necessarily lead to a mechanistic understanding [1]. During the last decade, metabolomics has emerged as a powerful approach in the microbiome field, providing functional information about the human gut microbial phenotype [7]. Therefore, a combined microbiome and metabolome strategy to evaluate host–microbiome interactions are being increasingly utilized.

To date, the focus of most metabolomics studies aimed at elucidating the role of the gut microbiome–metabolome co-axis has primarily been on water-soluble polar metabolites (e.g., tryptophan catabolites such as indole acetic acid and short-chain fatty acids (SCFA)). Other, nonpolar microbial metabolites, including lipids such as sphingolipids (SLs), endocannabinoids (ECs), cholesterol, bile acids, and acylcarnitines are less studied in comparison. However, lipids also have an important role in the gut microbiome–host interactions [8]. Gut microbiota not only regulate intestinal lipid absorption and metabolism but also impact levels and metabolism of a substantial proportion of circulating lipids [9,10]. Lipids are critical biomolecules involved in a wide range of cellular functions including structure, communication, and metabolism.

The lipidomic analysis of feces can identify numerous microbial lipids, which can inform about the gut microbial phenotype [11]. However, only a limited number of studies have integrated the lipidome in microbiome analyses with respect to health outcomes. In addition, the dynamics of microbial lipids in the gut are poorly understood. This could be ascribed to the fact that it is not feasible to perform dynamic sampling across the human gastrointestinal tract. To overcome this challenge, in vitro colon models have been extensively applied to study microbial functions [12]. Here we employed a lipidomics approach on in vitro-derived intestinal chyme to examine the temporal lipid changes occurring in different compartments of the colon simulator representing the proximal to the distal part of the colon. We also studied whether in vitro gut lipidome profiles were affected by colon simulation time and polydextrose (PDX), a synthetic complex oligosaccharide.

## 2. Materials and Methods

### 2.1. In Vitro Colon Simulator

The Enteromix model of the human large intestine (Figure 1) was described in detail previously [13,14]. In summary, each simulator unit consists of four connected glass vessels that are fed semi-continuously every third hour. The four vessels in the simulator (V1–V4) model different compartments of the human colon from the proximal (V1) to the distal part (V4), each having a different controlled pH and flow rate. The simulator is maintained anaerobically and at 37 °C. In the initial phase, the simulator is inoculated with preincubated fecal microbes from a fresh fecal sample, which forms the microbiota of the entire model. The microbes are incubated in an artificial ileal medium [15] that is composed based on the analysis of the ileal content from sudden death victims [16]. The same medium is used to feed the simulator during its running, and functions as a carrier for the polydextrose. In the present study, the fecal samples for inoculation were provided voluntarily by three healthy Finnish volunteers. One fecal sample from one volunteer was used to inoculate the entire simulator. Independent simulations were created by inoculating the simulator with a fecal sample from another volunteer. The study and all methods used in it were carried out in accordance with relevant guidelines and regulations, and informed consent was orally obtained from all research subjects. This simulation was performed at IFF, Kantvik, Finland. To understand the lipidomics changes over time, the microbial slurry was collected from all vessels (V1–V4) after 24 and 48 h with/without PDX treatment. Although gastrointestinal passage may be as short as 24 h, there is often a longer residence time in the intestine. Furthermore, due to the nature of the simulator being fed only every third hour, it takes time for the content to reach an equilibrium in the vessels, similar to the human colon. A total of 44 samples were gathered from vessels (V1–V4) of the in vitro colon simulator, and were kept at −80 °C until lipidomics analysis. These samples were obtained from 11 simulations, each involving four vessels. Among these simulations, eight were conducted for 48 h, while three were conducted for 24 h. In addition, media and inoculum were also used for the simulation and the pooled human fecal sample was collected as a quality control sample.

### 2.2. Lipidomics Analysis

Simulated intestinal chyme lipid extracts were prepared using a method based on the Folch procedure [17], as detailed by Lamichhane et al. [18]. An internal standard mixture containing 2.5 µg/mL 1,2-diheptadecanoyl-sn-glycero-3-phosphoethanolamine (PE(17:0/17:0)), N-heptadecanoyl-D-erythro-sphingosylphosphorylcholine (SM(d18:1/17:0)), N-heptadecanoyl-D-erythro-sphingosine (Cer(d18:1/17:0)), 1,2-diheptadecanoyl-sn-glycero-3-phosphocholine (PC(17:0/17:0)), 1-heptadecanoyl-2-hydroxy-sn-glycero-3-phosphocholine (LPC(17:0)), 1-palmitoyl-d31-2-oleoyl-sn-glycero-3-phosphocholine (PC(16:0/d31/18:1)) and 1,2,3-triheptadecanoyl-sn-glycerol (TG(17:0/17:0/17:0)) was prepared in CHCl_3_:MeOH (2:1, *v*/*v*). Six-point calibration curves with concentrations between 100 and 2500 ppb in CHCl_3_:MeOH (2:1, *v*/*v*) were prepared for 1-Hexadecanoyl-2-octadecanoyl-sn-glycero-3-oethanolamine (PE(16:0/18:1)), octadecenoyl-sn-glycero-3-phosphocholine (LPC(18:1)), cholesteryl hexadecanoate (CE(16:0)), 1,2-Distearoyl-sn-glycero-3-phosphoethanolamine (PE(18:0/18:0)), N-stearoyl-D-erythro-sphingosylphosphorylcholine (SM(18:0/18:1)) and cholesteryl linoleic acid (CE(18:2)). The samples were prepared by spiking 10 µL of the sample with 10 µL of 0.9% NaCl and 120 µL of internal standard solution. The samples were vortexed and were left to stand on ice for 30 min. Samples were centrifuged (9400× *g*, 5 min, 4 °C) and 60 µL from the lower layer was diluted with 60 µL of CHCl_3_:MeOH (2:1, *v*/*v*). For the liquid chromatography (LC) separation, a Bruker Elute UHPLC system (Bruker Daltonik, Bremen, Germany) equipped with an autosampler cooled to 10 °C, a column compartment heated to 50 °C and a binary pump was used. A Waters ACQUITY BEH C18 column (2.1 mm × 100 mm, 1.7 µm) was used for chromatographic separation. The flow rate was 0.4 mL/min and the injection volume was 1 µL. The needle was washed with 10% DCM in MeOH and ACN: MeOH: IPA: H_2_O (1:1:1:1, *v*/*v*/*v*/*v*) + 0.1% HCOOH after each injection for 7.5 s each. The eluents were H_2_O + 1% NH_4_Ac (1M) + 0.1% HCOOH (A) and ACN: IPA (1:1, *v/v*) + 1% NH_4_Ac + 0.1% HCOOH (B). The gradient was as follows: from 0 to 2 min, 35–80% B, from 2 to 7 min, 80–100% B, and from 7 to 14 min, 100% B. Each run was followed by a 7 min re-equilibration period under initial conditions (35% B).

Mass spectrometric detection was performed on a Bruker Impact II QTOF (Bruker Daltonik, Bremen, Germany). For data preprocessing, the raw data files were converted to a .mzml file using Bruker compass data analysis 5.1. The preprocessing was performed in MZmine2 (version 2.53) according to Thomas et al. [19]. Briefly, centroid mass detection was performed, followed by ADAP chromatogram builder, chromatogram deconvolution (local minimum search), and isotopic peaks grouper with join aligner. After this, a filtering step (feature list row filter), a custom database search, an adduct search, and gap filling (peak finder) were performed. Finally, the results were exported as a CSV file. After this, lipid class-based normalization was performed using the class-based internal standards; class-based calibration curves were created, and semi-quantification was performed using the calibration curves. Features that were annotated and had a relative standard deviation of less than 30% in the quality control samples were selected for further processing.

### 2.3. Endocannabinoid Analysis

The concentrations of endocannabinoids and endocannabinoid-like compounds were analyzed. Crash solvent (400 µL) consisting of acetonitrile (ACN), 0.1% formic acid (FA), and isotopically labeled internal standards (Appendix A) was added to a glass vial and 200 µL of chyme slurry was added. The samples were vortexed and left to settle at −20 °C for 30 min. The samples were filtered through protein precipitation filter plates and collected into 96-well plates with glass inserts. The samples were transferred to glass vials and dried at 35 °C under a gentle stream of nitrogen. The samples were reconstituted in 50 µL reconstitution solution (60% water, 20% ACN, and 20% isopropanol). The samples were analyzed using LC–MS. The chromatographic separation was performed on a Sciex exion (AB Sciex Inc., Framingham, MA, USA) consisting of a binary pump, an autosampler, and a thermostatic column compartment. The column used was an XBridge BEH C18 2.5 µm, 2.1 × 150 mm column with a precolumn made of the same material. The eluents were A: 0.1% FA and 1% ammonium acetate (1M) in water, and B: 0.1% FA and 1% ammonium acetate (1M) in ACN/IPA (50:50). The gradient is presented in Appendix A. The injection volume was 1 µL, the flow rate was 0.4 mL/min, and the column oven temperature was 40 °C. The detection was performed on a Sciex 7500 QTrap operating in MRM mode. The parameters used are presented in Appendix A. Quantification was performed using calibration curves from 0.01 ppb to 80 ppb (0.1 and 800 ppb for arachidonic acid (AA)) using the internal standard method. Quantification was performed with Sciex OS analytics. Due to the rapid isomerization of 2-AG to 1-AG, the results are presented as the concentration of total AG.

### 2.4. Data Analysis

Lipid data values were log-transformed prior to multivariate analysis. The difference in the lipidome between the different vessels, time, and case (with/without PDX treatment) were analyzed using a multivariate linear model using the MaAsLin2 package in R (lipids ∼ Time + Vessel + case). Adjusted *p*-values of 0.25 were considered significant. Spearman correlation coefficients were calculated using the Statistical Toolbox in MATLAB 2017b and *p*-values < 0.05 (two-tailed) were considered significant for the correlations. The individual Spearman correlation coefficients (R) were illustrated as a heat map using the ‘‘corrplot’’ package (version 0.84) for the R statistical programming language.

## 3. Results

### 3.1. Untargeted Lipidomics and Targeted Endocannabinoid Analysis in the Simulated Fecal Samples

We analyzed simulated intestinal chyme lipids obtained from different vessels (V1–V4) in the in vitro colon, which mimics the compartments of the human colon from the proximal to the distal part (Figure 1). In total, 44 simulated intestinal chyme samples were collected from the in vitro colon simulator (vessels V1–V4) and ran for either 24 or 48 h with/without PDX treatment. The untargeted lipidomics assay of the simulated chyme extract resulted in the detection of 118 annotated lipids. These lipids were semi-quantified using class-specific internal standards and calibration curves. The semi-quantified lipids included a wide range of lipid classes: triacylglycerols (TG), ceramides (Cer), cholesterol esters (CE), diacylglycerols (DG), lysophosphatidylcholines (LPC), phosphatidylcholines (PC), phosphatidylethanolamines (PE), and sphingomyelins (SM). Of the 13 endocannabinoids (EC) studied, 11 were detected in at least one sample type. These included palmitoylethanolamide (PEA), arachidonoyl glycerol (AG), 2-arachidonic glycerol ether (2-Age), arachidonoylethanolamide (AEA), oleoylethanolamide (OEA), stearoylethanolamide (SEA), docosatetraenoylethanolamide (DEA), alpha-linolenoylethanolamide (aLEA), arachidonic acid (AA), N-arachidonoyl taurine (NAT) and N-arachidonoyl-L-serine (NALS). Nine (PEA, AG, 2-AGe, AEA, OEA, SEA, DEA, aLEA, and AA) were detected in human feces, which were used as quality control samples.

### 3.2. Lipidome in the In Vitro Colon Simulator

Principal component analysis (PCA) of the preprocessed lipidomics data revealed a clear vessel-related pattern in the simulated intestinal chyme samples. To examine the contributions of various factors to simulated chyme lipidome profiles, multivariate linear modeling was performed (lipids ∼ Time + Vessel + treatment). We found that the vessel had a marked impact on the simulated chyme lipidome when compared to the simulation time (24 vs. 48 h) and treatment (with/without PDX treatment). Of the analyzed lipids, 40 showed a significant change in at least one of the vessels (*p* < 0.05, Figure 2A and Appendix A Appendix A). These lipids included one CE, two DGs, five Cer, eleven PCs, one PG, four PEs, four SMs, and nine TGs (Figure 2A–C). All of these lipids passed the FDR threshold of 0.25 (Appendix A). With the exception of Cer(d18:1/20:0), LPE (16:0), and LysoPE(18:1), most of the lipids showed a decreased pattern from vessel V1 to V4, i.e., from the proximal to the distal part of the colon simulator (Figure 2B–D). Among the TGs, specifically, those TGs with a low double bond count (≤2 double bonds) showed changes within the vessels (V1–V4, Appendix A). However, no clear pattern with respect to the double bond counts and/or carbon number compositions was observed in any other specific lipid class.

We also found that chyme lipidome concentrations of 26 lipids, mainly TGs (n = 14), were different across two time points (24 vs. 48 h of simulation, Appendix A), while only four lipids (DG(34:2), LacCer(d18:1/18:0), TG (55:6)/TG (16:0/19:1/20:5), and TG (51:1)) were found altered when the simulation was performed with/without PDX.

Next, we analyzed the dynamics of ECs in the different compartments of the colon simulator over time. Among the 11 detected ECs, the levels of 7 ECs were altered in at least one of the vessels in the colon simulator (*p* < 0.05, Figure 3A and Appendix A). These ECs include AEA, aLEA, DEA, OEA, PEA, and SEA. There was no persistent trend; however, the levels of OEA, aLEA, and PEA increased in vessels (V3–V4) when compared to vessel V1 (Figure 3B). A similar trend was seen for AEA with a higher level of variation appearing in vessel V3 (Figure 3C). On the other hand, the level of SEA was higher in V1 when compared to vessels V2-V4 (Figure 3 and Appendix A). In addition, we analyzed the EC concentrations over time in the simulated gut. Overall, the levels of five ECs (AEA, 2-AGe, AA, PEA, and NAT) were increased in the colon simulation when ran for 48 h compared to 24 h for the intestinal chyme slurry (Appendix A).

Given the known link between gut microbiota and EC metabolism [20], we also examined the difference in ECs profiles between the media used for in vitro gut simulation and the simulated chyme slurry. We found most of the ECs detected in the simulated chyme slurry were lower in concentration than in the simulation media. Interestingly, we observed NALS was detected at low concentrations in the media; however, it was not detected in any of the vessels (Figure 4A). Meanwhile, 2-AGe was not detected in the media but it appeared in different vessels (Figure 4B, V1–V3).

### 3.3. Association of Lipidome and ECs in the In Vitro Colon Simulator

Next, we performed a correlation analysis between ECs and individual simulated intestinal chyme lipid levels (Figure 5). We found that the levels of SEA and DEA were positively associated with the overall simulated chyme lipidome. OEA, aLEA, and PEA showed a positive association with Cers and LPEs, while those being inversely correlated were PCs, PEs, SMs, and TGs. This trend was less pronounced for AA and AG. Instead, there was a clear inverse trend association between SMs and PEs/AA levels in the in vitro simulator. In addition, DGs were negatively related to the level of AA and AG. No association between individual lipids and NAT was observed, except for Cer(d18:1/20:0).

## 4. Discussion

In this study, we reported the dynamics of lipids in each compartment of the colon simulator. We found a more distinct lipids profile in the proximal colon (vessel V1) than in the distal part of the colon (vessel V4). Specifically, specific Cer, PCs, SMs, and TGs were decreased in the distal as compared to the proximal part of the colon simulator. Our observations also showed that in vitro-derived intestinal chyme lipids, particularly TGs, are strongly affected by time. Our results are in agreement with previous studies, showing that profound metabolic changes occur in different parts of the in vitro gut over time [12]. The level of SCFAs (acetate, butyrate, and propionate), branched-chain fatty acids (iso-valerate), biogenic amines (trimethylamine), organic metabolites (succinate, ethanol, formate, valerate, and n-acetyl compounds) and amino acids (lysine, leucine, isoleucine, phenylalanine, tyrosine, and valine) were reported to change in the four vessels (V1–V4) within 48 h (12, 24, 36, and 48 h) [12,14]. Similar dynamic changes in the metabolome along the intestinal tract have been reported in nonhuman primates [21].

The human gut is an endogenous source of systemic sphingolipids [22]. We observed distinct changes in the levels of the sphingolipids (Cer and SMs) while passing from vessel V1 to V4 over time (24 and 48 h). Sphingolipids were higher in the proximal colon (V1) than in the distal part of the colon (V4). Sphingolipids have many structural and signaling roles in eukaryotes [22], and microbially-derived sphingolipids may markedly impact the host sphingolipid levels [23,24]. In addition, sphingolipids have been shown to promote the survival of commensal bacteria [25]. Notably, the observed decreasing trend in our study may indicate increased microbial catabolism. Lipid is considered an alternative pathway for carbon, nitrogen, and an energy source for the gut microbes [26]. We acknowledge the lack of microbiome data as the main limitation of our study; therefore, we could not demonstrate a causal link between gut microbiota and sphingolipid metabolism. Notwithstanding this, our study introduced a simulated intestinal chyme lipidomics-based method to interrogate potential microbial lipids within the complex gut system. Our study also highlights that an in vitro colon model provides a controlled experimental condition, i.e., dynamic sampling for the characterization of gut microbiota-derived bioactive lipids. Thus, it provides a potentially significant strategy for understanding the intricate relationship between gut microbes and lipids, which may open new avenues to studying human metabolism.

Growing evidence suggests that bidirectional interactions exist between gut microbes and the endocannabinoid system [27,28,29,30]. The endocannabinoid system comprises a network of cannabinoid-type receptors and their ligands (i.e., the endocannabinoids) that are present throughout the human body. The concept of the endocannabinoid system and the gastrointestinal tract is over a half-century old. Studies from the 1970s have already shown that ECs can have a profound impact on gut motility [20,31,32]. Current evidence suggests that gut transit times affect gut microbial composition and function [33]. Here, we found that the concentration of ECs varied between the proximal (V1) and distal colon (V4). We also found an endocannabinoid of potential microbial origin (i.e., 2-AGe). Phospholipids are precursors of ECs [34]. Intriguingly, we found an inverse association between PCs and those ECs that were increasing from V1 to V4. We thus hypothesize that the availability of PCs as a substrate in the gut drives the level of specific ECs. In addition, this finding highlights the role of microbes in the metabolism of ECs in the gut.

## 5. Conclusions

In summary, our study shows that the combination of lipidomics and in vitro-derived intestinal chyme samples enabled us to characterize the fate of lipids in a simulated human colon. Our study also reports a novel profile of ECs in the simulated intestinal chyme.

## Figures and Tables

**Figure 1 metabolites-13-00355-f001:**
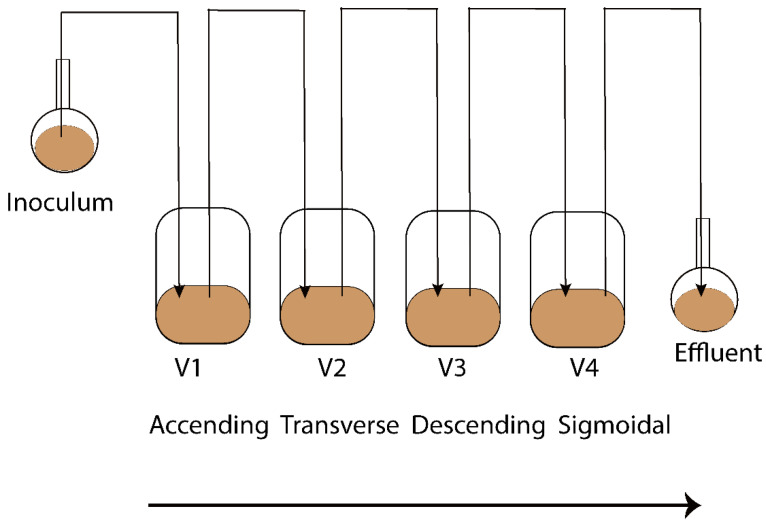
Schematic representation of the Enteromix simulated colon model. The vessels in one unit (V1–V4) simulate the different compartments of the human colon from the proximal to the distal part, each having a different controlled pH and flow rate. The whole unit is maintained anaerobically and at 37 °C. See [13] for more details. A total of 44 samples were gathered from vessels (V1–V4) of the in vitro colon simulator and were kept at −80 °C until lipidomics analysis. These samples were obtained from 11 simulations, each involving four vessels. Among these simulations, eight were conducted for 48 h, while three were conducted for 24 h.

**Figure 2 metabolites-13-00355-f002:**
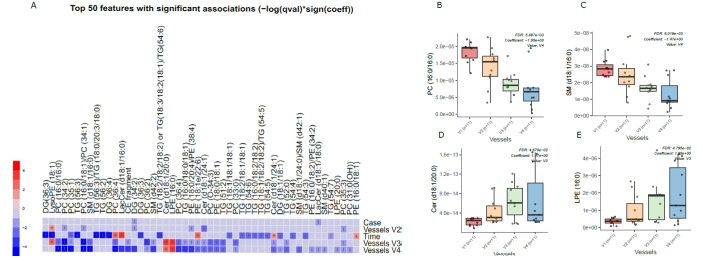
Dynamics of lipids in the in vitro colon simulator. The figure depicts the lipidomic changes in four vessels (V1–V4) over two time points (24 and 48 h). (**A**) Heat map showing the difference in the lipidome between vessels (V1–V4), time (24 vs. 48 h), and case (with/without PDX) compared using a multivariable linear model. The changes in the vessels were referenced to vessel 1. Blue indicates decreasing trend while red denotes an increase in the trend. (**B**–**E**) The box plot shows the difference in the lipidome between vessels (V1–V4). These selected lipids highlight the trend between the vessels (V1–V4), which mimics the compartments of the human colon from the proximal to the distal part.

**Figure 3 metabolites-13-00355-f003:**
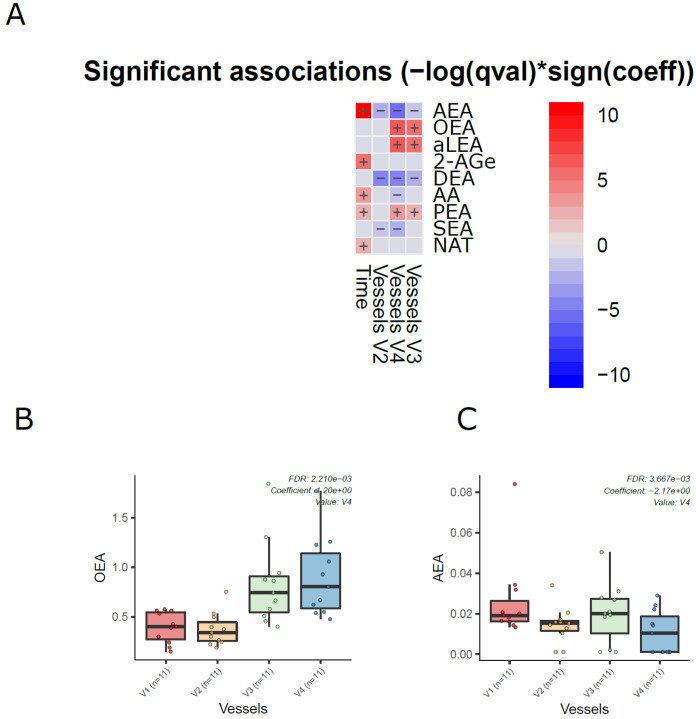
Dynamics of ECs in the in vitro colon simulator. The figure depicts the EC changes in four vessels (V1–V4) over two time points (24 and 48 h). (**A**) Heat map showing the difference in the EC between vessels (V1–V4), time (24 vs. 48 h), and case (with/without PDX) compared using a multivariable linear model. The changes in vessels were in reference to vessel 1. Blue indicates decreasing trend while red denotes an increase in the trend. (**B**,**C**) The box plot shows the difference in the EC between vessels (V1–V4). These selected ECs highlight the dynamic trends between the vessels (V1 to V4), which mimics the compartments of the human colon from the proximal to the distal part.

**Figure 4 metabolites-13-00355-f004:**
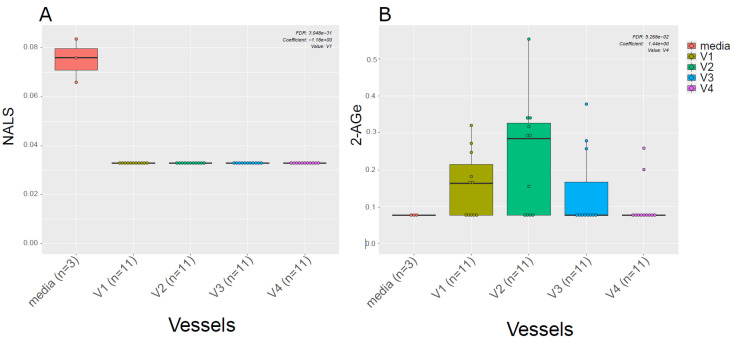
ECs in the media vs. in vitro colon simulator. The figure depicts the endocannabinoid changes in four vessels (V1–V4) over two time points (24 and 48 h). (**A**,**B**) The box plot shows the difference in the EC between vessels (V1–V4) and media used during the simulation.

**Figure 5 metabolites-13-00355-f005:**
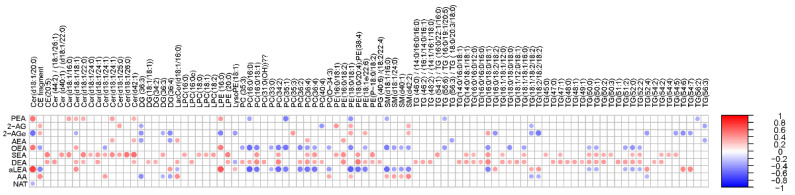
Association of the lipidome and ECs in the in vitro colon simulator. Spearman correlation coefficients illustrated by heat map.

## Data Availability

The lipidomics datasets generated in this study will be submitted to the Metabolomics Workbench repository (https://www.metabolomicsworkbench.org).

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
