# Peer review of "Dynamics of the Lipidome in a Colon Simulator"

_metabolites, 2023, doi:10.3390/metabo13030355_

Round 1

Reviewer 1 Report

This paper describes an interesting study investigating changes in the lipidome within a colon simulator. The paper is well-written, and I enjoyed learning about the system. Here are my comments:

·        The in vitro colon simulator is not adequately described. The authors indicated it had been published elsewhere, but more detail needs to be provided in this paper to understand what was done. For example, line 63 mentioned eight separate units. It is not clear why that is important. Couldn’t it be any number? The description sounds like each unit contains four vessels that feed into each other sequentially. To me, this configuration (Figure 1) is the essence of the system, not how many units there are. Perhaps I don’t understand.

·        Fecal samples are collected from three Finnish volunteers. Are these samples mixed, or was each one studied separately?

·        Did each volunteer provide a fecal sample from each area of their colon (ascending, transverse, descending, and sigmoid)? Or was it a dedicated fecal sample?

·        Are the fecal samples mixed with anything to keep the microbes growing? Or were they mixed with chyme? What was this composed of, and wouldn’t that impact the lipidome?

·        Why were 24 and 48 hours the time points? Food transits the human digestive tract in 24 hours. So essentially, the study showed that time, pH, and flow rate could impact the lipids and EC produced by gut microbes. Wouldn’t that be expected?

·        Forty-four samples were obtained. I can’t figure out the math and when these were collected. If there are four vessels and two-time points, that would be eight distinct collection points. Eight does not go into 44 evenly.

·        Figure 1 indicates an inoculum. What was this? Just the fecal samples combined? This mixture would not reflect the chyme dumped from the small intestine into the large intestine.

·        Figure 1: Inoculum is not correctly spelled.

·        Line 116: I believe a version number is missing.

·        Line 117: I believe a reference is missing.

·        Line 118-119: Is “local minimum search” correct?

·        Line 120: Is “Peak finder” correct?

·        Figure 2, Figure 3B, Figure 3C, Figure 4, and Figure 5 are too small to read.

·        Figure 2B: What is B? I don’t see that included in the legend.

·        Figure 4A: Is the value zero for each of the vessels, or is it very small (almost zero)? The y scale could be modified to allow the vessel values to be bigger.

·        Although the authors state they do not have the microbiota data, it would still be interesting to include a paragraph on this topic. Can the authors postulate that the values obtained might reflect the human gut?

·        A paragraph on weaknesses should be included in the discussion.

Author Response

Reviewer 1

This paper describes an interesting study investigating changes in the lipidome within a colon simulator. The paper is well-written, and I enjoyed learning about the system. Here are my comments:

The in vitro colon simulator is not adequately described. The authors indicated it had been published elsewhere, but more detail needs to be provided in this paper to understand what was done. For example, line 63 mentioned eight separate units. It is not clear why that is important. Couldn’t it be any number? The description sounds like each unit contains four vessels that feed into each other sequentially. To me, this configuration (Figure 1) is the essence of the system, not how many units there are. Perhaps I don’t understand.

Response: We have rephrased the section to better indicate how one simulator unit is operated. Indeed, the number of units is, in practice eight, but could be any number. The edited text is included below:

“The Enteromix model of the human large intestine (Fig. 1) was described in detail previously [13, 14]. In summary, each simulator unit consists of four connected glass vessels that are fed semi-continuously every third hour. The four vessels in the simulator (V1–V4) model different compartments of the human colon from the proximal (V1) to the distal part (V4), each having a different controlled pH and flow rate. The simulator is kept anaerobically and at 37 °C. In the initial phase, the simulator is inoculated with pre-incubated fecal microbes from a fresh fecal sample, which form the microbiota of the entire model. The microbes are incubated in artificial ileal medium (G. T. Macfarlane et al., 1998) that is composed based on analysis of ileal content from sudden death victims (G.T. Macfarlane et al., 1992). The same medium is used to feed the simulator during its running and functions as a carrier for the polydextrose. In the present study, the fecal samples for inoculation were provided voluntarily by three healthy Finnish volunteers. Each fecal sample was used separately to create independent simulations. The study and all methods used in it were carried out in accordance with relevant guidelines and regulations, and informed consent was orally obtained from all research subjects. This simulation was performed at IFF, Kantvik, Finland. To understand the lipidomic changes over time, the microbial slurry was collected from all vessels (V1–V4) after 24 and 48 h with/without PDX treatment. Although gastrointestinal passage may be as short as 24h there is often a longer residence time in the intestine. Furthermore, due to the nature of the simulator being fed only every third hour, it takes time for the content to reach an equilibrium in the vessels, similar as in the human colon. In total 44 samples were collected from the in vitro colon simulator vessels (V1–V4) and stored at −80 °C prior to lipidomic analysis. In addition, also media and inoculum used for the simulation and the pooled human fecal sample as a quality control sample were collected.”

Fecal samples are collected from three Finnish volunteers. Are these samples mixed, or was each one studied separately?

Response: The fecal samples were studied separately to have independent simulations. This has been better clarified in the text (see above).

Did each volunteer provide a fecal sample from each area of their colon (ascending, transverse, descending, and sigmoid)? Or was it a dedicated fecal sample?

Response: One fecal sample was used to inoculate the entire model. This has been clarified in the text (see above).

Are the fecal samples mixed with anything to keep the microbes growing? Or were they mixed with chyme? What was this composed of, and wouldn’t that impact the lipidome?

Response: The following text was added to the manuscript:

“The microbes are mixed with artificial ileal chyme. The chyme was formulated on the composition of the small intestinal contents from sudden death victims (G.T. Macfarlane, Gibson, Beatty, & Cummings, 1992). The composition of the artificial chyme is as follows: in distilled water (g/l): starch (BDH Ltd.), 5.0; peptone, 0.05; tryptone, 5.0; yeast extract, 5.0; NaCl, 4.5; KCl, 4.5; mucin (porcine gastric type III), 4.0; casein (BDH Ltd.), 3.0; pectin (citrus), 2.0; xylan (oatspelt), 2.0; arabinogalactan (larch wood), 2.0; NaHCO3, 1.5; MgSO4, 1.25; guar gum, 1.0; inulin, 1.0; cysteine, 0.8; KH2PO4, 0.5; K2HPO4, 0.5; bile salts No. 3, 0.4; CaCl2 ×6H2O, 0.15; FeSO4 ×7H2O, 0.005; hemin, 0.05; and Tween 80, 1.0. (G. T. Macfarlane, Macfarlane, & Gibson, 1998). We have clarified the use of the artificial chyme but do not mention the exact composition of it. Instead we provide the reference to it.”

Macfarlane, G. T., Gibson, G. R., Beatty, E., & Cummings, J. H. (1992). Estimation of short-chain fatty acid production from protein by human intestinal bacteria based on branched-chain fatty acid measurements. FEMS Microbiol. Lett., 101(2), 81-88. doi:https://doi.org/10.1016/0378-1097(92)90829-D

Macfarlane, G. T., Macfarlane, S., & Gibson, G. R. (1998). Validation of a Three-Stage Compound Continuous Culture System for Investigating the Effect of Retention Time on the Ecology and Metabolism of Bacteria in the Human Colon. Microb.Ecol., 35(2), 180-187. Retrieved from PM:9541554

Why were 24 and 48 hours the time points? Food transits the human digestive tract in 24 hours. So essentially, the study showed that time, pH, and flow rate could impact the lipids and EC produced by gut microbes. Wouldn’t that be expected?

Response: The transit time of food varies between individuals; 24h is maybe ‘optimal’. We have chosen here 24 and 48h as is takes time for the microbial inoculum and the ‘food’ i.e. polydextrose to travel through the system and reach some sort of equilibrium as exists in the colon. We have explained this more carefully in the text.

Forty-four samples were obtained. I can’t figure out the math and when these were collected. If there are four vessels and two-time points, that would be eight distinct collection points. Eight does not go into 44 evenly.

Response: We collected samples from 11 simulations (n = 44, come from11*4 i.e. 4 vessels), of which 8 were simulated for 48 h and 3 were for 24 h. This information has been clarified in the revised version of the manuscript.

Figure 1 indicates an inoculum. What was this? Just the fecal samples combined? This mixture would not reflect the chyme dumped from the small intestine into the large intestine.

Response: Fecal sample from the donor was used to inoculate the colon model. This information has been clarified in the revised manuscript.

Figure 1: Inoculum is not correctly spelled.

Response: We have corrected this error and updated the manuscript.

Line 116: I believe a version number is missing.

Response: The version was added.

Line 117: I believe a reference is missing.

Response: The correct reference was added.

Line 118-119: Is “local minimum search” correct?

Response: That is the name of the algorithm used in mzmine2.

Line 120: Is “Peak finder” correct?

Response: That is the name of the algorithm used in mzmine2.

Figure 2, Figure 3B, Figure 3C, Figure 4, and Figure 5 are too small to read.

Response: Figures are uploaded as vector image for further clarity.

Figure 2B: What is B? I don’t see that included in the legend.

Response: This information has been clarified in the revised manuscript.

Figure 4A: Is the value zero for each of the vessels, or is it very small (almost zero)? The y scale could be modified to allow the vessel values to be bigger.

Response: Figure 4 has been modified as per the context suggested by the reviewer. NALS were not detected in the simulated intestinal chime, on the other hand 2AGe were only detected in the simulated chime. The values are close to zero as they are imputed with half of the minimum of non-missing elements.

Although the authors state they do not have the microbiota data, it would still be interesting to include a paragraph on this topic. Can the authors postulate that the values obtained might reflect the human gut? A paragraph on weaknesses should be included in the discussion.

Response: We agree with the reviewer. The discussion section is now updated as per the context suggested by the reviewer.

Reviewer 2 Report

Missing a few things:

Line 114: Not filled - Bruker Impact II QTOF (manufacturer, city, country);

Line 116: Bruker Compass version no;

Line 117: Reference for MZmine (I would also suggest using the correct capitalisation in the text = MZmine 2)

I think the section “5. Conclusions” is missing… Or maybe it is just not labelled …

Author Response

Reviewer 2

Missing a few things:

Line 114: Not filled - Bruker Impact II QTOF (manufacturer, city, country);

Response: This information was added to the text.

Line 116: Bruker Compass version no;

Response: This information was added to the text.

Line 117: Reference for MZmine (I would also suggest using the correct capitalisation in the text = MZmine 2)

Response: This was changed in the text.

I think the section “5. Conclusions” is missing… Or maybe it is just not labelled …

Response:  This information has been added in the revised manuscript.

Reviewer 3 Report

The manuscript of Kråkström et al. represents a new and very clinically relevant direction - lipidomic analysis of feces. The work would look even more interesting if the authors paid more attention to the role of individual fecal lipids in physiology or pathophysiology. Despite the importance of the direction, the form in which the work is presented rather resembles a technical sketch of a future in-depth article. In the absence of the role of the determined lipids, the clinical significance of the results is unclear. Also noteworthy is the absence of significant differences in lipid levels in different parts of the colon. A trend is not a change. Also puzzling is the authors' statement that "adjusted p values with FDR = 0.25 were considered significant."

In addition to the conceptual recommendation, there are a couple of minor drawbacks:

·         all abbreviations must be explained, including PDX;

·         of the 13 defined "endocannabinoids", most do not belong to this class, since they do not activate cannabinoid receptors, but produce endocannabinoid-like effects;

·         what are the criteria for a healthy fecal microbiota and how different was the microbiota in the three volunteers? Is it possible that the absence of significant differences in the final lipid content is the result of differences in the microbiota in volunteers?

Author Response

Reviewer 3

The manuscript of Kråkström et al. represents a new and very clinically relevant direction - lipidomic analysis of feces. The work would look even more interesting if the authors paid more attention to the role of individual fecal lipids in physiology or pathophysiology. Despite the importance of the direction, the form in which the work is presented rather resembles a technical sketch of a future in-depth article. In the absence of the role of the determined lipids, the clinical significance of the results is unclear. Also noteworthy is the absence of significant differences in lipid levels in different parts of the colon. A trend is not a change. Also puzzling is the authors' statement that "adjusted p values with FDR = 0.25 were considered significant."

Response: As suggested by the reviewer we have highlighted the role of relevant fecal lipids, for instance sphingolipids in the context of host physiology or pathophysiology in the discussion section of the manuscript. However, at the same time we also acknowledge that to establish a meaningful association between lipids and phenotype systematic integration of numerous host factors are required, especially to draw strong conclusions regarding cause and effect relationships. Therefore, that is highlighted as a limitation of our study. Next, we agree with the reviewer that trend is not changes and clarified the significance statement highlighted by the reviewer.

In addition to the conceptual recommendation, there are a couple of minor drawbacks:

all abbreviations must be explained, including PDX

Response: The abbreviation PDX is explained on line 58 and the abbreviation in the abstract was changed to polydextrose. The manuscript was reviewed for abbreviations with missing explanations.

of the 13 defined "endocannabinoids", most do not belong to this class, since they do not activate cannabinoid receptors, but produce endocannabinoid-like effects;

Response: This information has been modified in the revised manuscript. The following sentence was added: “The concentrations of endocannabinoids and endocannabinoid-like compounds was analysed.“

what are the criteria for a healthy fecal microbiota and how different was the microbiota in the three volunteers? Is it possible that the absence of significant differences in the final lipid content is the result of differences in the microbiota in volunteers?

Response: We don’t have data on the fecal microbiota. We have acknowledged it as a limitation in the discussion section of the revised manuscript. We also agree with the reviewer, lipid content may be the result of differences in microbiome content, which is highlighted in the discussion section of the manuscript.

Round 2

Reviewer 1 Report

The authors have made modifications to their manuscript, as suggested. I appreciate their efforts in doing that. However, I still have a few concerns.

1)     The authors have improved the description of the system; however, the methods paragraph states, “In the present study, the fecal samples for inoculation were provided voluntarily by three healthy Finnish volunteers.” In their response to my specific question about this, the authors state, “One fecal sample was used to inoculate the entire model.” I am still confused about what was done.

2)     Information to explain the 44 was provided, but I still don’t understand. I get the 11 simulations * 4 vessels equals 44. But what are the ”8 were simulated for 48 h and 3 were for 24 h?”

The heat map in Figure 2 is not readable.

The other additions and changes are acceptable.

Author Response

1)     The authors have improved the description of the system; however, the methods paragraph states, “In the present study, the fecal samples for inoculation were provided voluntarily by three healthy Finnish volunteers.” In their response to my specific question about this, the authors state, “One fecal sample was used to inoculate the entire model.” I am still confused about what was done.

RESPONSE: We apologize for the ambiguity of the formulation. We hope the following formulation is clear: "In the present study, the fecal samples for inoculation were provided voluntarily by three healthy Finnish volunteers. One fecal sample from one volunteer was used to inoculate the entire simulator. Independent simulations were created by inoculating the simulator with a fecal sample from another volunteer." This has been updated in the revised manuscript.

2)     Information to explain the 44 was provided, but I still don’t understand. I get the 11 simulations * 4 vessels equals 44. But what are the ”8 were simulated for 48 h and 3 were for 24 h?”

RESPONSE: A total of 44 samples were gathered from vessels (V1-V4) of the in vitro colon simulator, and were kept at -80°C until lipidomics analysis. These samples were obtained from 11 simulations, involving four vessels each. Among these simulations, eight were carried out for 48 hours, while three were conducted for 24 hours. This has been updated in the revised manuscript.

The heat map in Figure 2 is not readable.

RESPONSE: The orientation of the heatmap has been changed to enhance readability.